# Breaking Barriers: Bioinspired Strategies for Targeted Neuronal Delivery to the Central Nervous System

**DOI:** 10.3390/pharmaceutics12020192

**Published:** 2020-02-23

**Authors:** Ana P. Spencer, Marília Torrado, Beatriz Custódio, Sara C. Silva-Reis, Sofia D. Santos, Victoria Leiro, Ana P. Pêgo

**Affiliations:** 1i3S—Instituto de Investigação e Inovação em Saúde, Universidade do Porto, Rua Alfredo Allen 208, 4200-135 Porto, Portugal; ana.spencer@i3s.up.pt (A.P.S.); marilia.torrado@i3s.up.pt (M.T.); beatriz.custodio@i3s.up.pt (B.C.); sara.reis@i3s.up.pt (S.C.S.-R.); sasantos@ineb.up.pt (S.D.S.); victoria.leiro@ineb.up.pt (V.L.); 2INEB—Instituto de Engenharia Biomédica, Universidade do Porto, Rua Alfredo Allen 208, 4200-135 Porto, Portugal; 3FEUP—Faculdade de Engenharia, Universidade do Porto, 4200-465 Porto, Portugal; 4ICBAS—Instituto de Ciências Biomédicas Abel Salazar, Universidade do Porto, 4050-313 Porto, Portugal

**Keywords:** CNS disorders, drug delivery, nanomedicine, nanocarriers, bioinspired, neurotargeting

## Abstract

Central nervous system (CNS) disorders encompass a vast spectrum of pathological conditions and represent a growing concern worldwide. Despite the high social and clinical interest in trying to solve these pathologies, there are many challenges to bridge in order to achieve an effective therapy. One of the main obstacles to advancements in this field that has hampered many of the therapeutic strategies proposed to date is the presence of the CNS barriers that restrict the access to the brain. However, adequate brain biodistribution and neuronal cells specific accumulation in the targeted site also represent major hurdles to the attainment of a successful CNS treatment. Over the last few years, nanotechnology has taken a step forward towards the development of therapeutics in neurologic diseases and different approaches have been developed to surpass these obstacles. The versatility of the designed nanocarriers in terms of physical and chemical properties, and the possibility to functionalize them with specific moieties, have resulted in improved neurotargeted delivery profiles. With the concomitant progress in biology research, many of these strategies have been inspired by nature and have taken advantage of physiological processes to achieve brain delivery. Here, the different nanosystems and targeting moieties used to achieve a neuronal delivery reported in the open literature are comprehensively reviewed and critically discussed, with emphasis on the most recent bioinspired advances in the field. Finally, we express our view on the paramount challenges in targeted neuronal delivery that need to be overcome for these promising therapeutics to move from the bench to the bedside.

## 1. The Need for New Neurotargeted Therapies

Neurological disorders are one of the major causes of death (9 million in 2016) and the leading cause of disability-adjusted life years (276 million in 2016) worldwide [1]. The number of people suffering from central nervous system (CNS) disorders has increased dramatically in the last few decades due to the population growing and aging, placing a tremendous burden on families and social economies [1]. CNS disorders encompass a huge number of pathological conditions induced by injuries or diseases, such as stroke, primary and metastatic tumors of the CNS, CNS infections, traumatic brain and spinal cord injuries, Alzheimer’s and Parkinson’s diseases, and multiple sclerosis, among others [2,3].

Despite this alarming scenario, there is a lack of effective therapies for the great majority of CNS disorders. Many efforts have been made in exploring approaches aimed at minimizing the clinical outcome caused by neuronal degeneration and death; however, the potential benefits observed in pre-clinical research contrasts with the failure when translated to the clinics. In the vast majority of these strategies, this has been mainly attributed to poor blood stability, limited tissue permeation before reaching the neuronal cells and, ultimately, low bioavailability at the CNS tissue of the investigated therapeutic agents, as well as their inadequate biocompatibility or peripheral side-effects [4,5]. Therefore, there is an imperative necessity for the development of new approaches that comply with clinical requirements.

With the progress of the nanotechnology field, one expects to observe great advances in CNS therapeutics, since nanoscaled biomaterials are anticipated to serve as promising delivery platforms (nanocarriers) for therapeutic (bio)molecules, being able to overcome the referred caveats. Besides the tunable characteristics (size, shape, surface properties, hydrophobicity, drug encapsulation or conjugation capacity and releasing), most of the known nanocarriers are able to be further functionalized with targeting moieties to improve their efficacy in reaching the tissue and/or cells of interest, increasing the bioavailability and delivery efficiency at the target site [6].

Indeed, the development of effective CNS therapies has been hampered by the lack of the ‘magic bullet’, able to specifically target the pathological area. Consequently, major efforts have been taken in the development of nanotechnology strategies with targeting properties to increase the bioavailability and delivery efficiency at the specific CNS or neuronal site.

In any administration of a therapeutic agent aimed at the CNS, there is the challenge in reaching the tissue first and neurons or other cell types later, as very few molecules have the ability to cross the CNS barriers. Among these barriers, we can enumerate the blood–brain barrier (BBB), the blood–cerebrospinal fluid barrier (BCSFB), the blood–spinal cord barrier (BSCB) and the avascular arachnoid barrier (AAB) [7,8], which will be further described in Section 4.2. Limitations in CNS barrier permeation inspired researchers in applying biological-inspired molecules to target CNS tissue or cells. In this review, we will present some of the developed nanobiomedical approaches, with particular focus on the bioinspired neuronal-targeted ones, as well as the main challenges along the delivery path that these therapeutic strategies may find.

## 2. The Nanotechnological Answer to an Efficient Therapy

Nanomedicine is the application of nanotechnology to biomedical applications, including diagnosis and drug delivery purposes, in a living organism. Drugs with very poor water solubility present several limitations, including limited transport and bioaccessibility after administration, and thus a higher quantity is often required so that the clinical effect can be attained, often at the cost of undesirable side effects. Other therapeutic agents, while soluble, may be rapidly degraded when administered, rapidly excreted and/or prevented by the different biological barriers to come into contact with the target site. In recent years, one has seen that many of these limitations could be circumvented by the introduction of nanotechnological-based delivery systems [9]. Over the past few decades, a variety of viral and non-viral nanocarriers have been extensively investigated to improve the clinical performance of several treatments. By exploring different delivery systems, the encapsulation/immobilization of drugs has improved their bioavailability and subsequent therapeutic efficacy, by increasing their solubility, protection from degradation in biological fluids and circulation times.

### 2.1. Nanosystems for Therapeutics Delivery

#### 2.1.1. Viral Delivery Systems

Viral vectors are constructed based on viruses, since they possess the natural ability to evade the host cells, delivering and exploiting the cellular machinery to express its own genetic material. The ideal viral carrier would be obtained by maintaining all the internalization, tropism and delivery capabilities of the wildtype virus, as well as its nanometric size (typically bellow 100 nm) but not carrying viral payloads, which could result in toxicity and an immunogenic response. This can be achieved by genetic engineering through the removal of the viral components responsible for the pathogenicity, while leaving intact all the necessary components for the payload delivery [10,11].

Several types of virus, including retrovirus, adenovirus, adeno-associated virus, and herpes simplex virus, have been explored [12,13]. Inspired by these viruses, various drug delivery systems, such as virus-like particles and virosomes have been proposed [11,14]. However, over the years, the use of these vectors has revealed some drawbacks, such as their difficult and expensive production in large quantities and the presence of residual viral elements that can potentially cause an inflammatory response, cytotoxicity, immunogenicity and insertional mutagenesis [12,14]. Thus, several efforts are being made to overcome these shortcomings. For further reading on virus-inspired nanocarriers for drug delivery, please refer to the review by Sabu et al. [14].

The disadvantages associated with viral vectors have led scientists to look for safer alternatives [5]. The development of non-viral nanosystems has glaring safety advantages as they present limited toxic and immunological problems. Moreover, non-viral delivery systems are easier to produce on a large scale through innovative synthesis schemes and they are not limited by the payload capacity of the virus capsule and, therefore, can transfer higher payloads. However, despite the great features of this class of vectors, when applied for gene delivery, the results of transfection are still far from the delivery efficiency obtained with viral vectors. Therefore, several research teams are working hard towards the improvement of non-viral vector design to enhance transfection efficiencies.

The four main types of non-viral delivery systems that have been explored for the delivery of therapeutics are discussed in the following sections: inorganic-, lipid-, polymer-, and dendrimer-based vectors (Figure 1).

#### 2.1.2. Non-Viral Delivery Systems

##### Inorganic Nanosystems

The field of inorganic-based nanosystems has exploded within recent decades. Silica, gold, silver and iron oxide nanoparticles (NPs) are among the inorganic nanosystems being explored for drug delivery. Currently, the porosity, shape and size of inorganic NPs can be precisely tuned. Additionally, although not exclusively characteristic of this type of NP, their surface can be easily modified.

Gold (Au) NPs present characteristics that have aroused the interest of the scientific community, such as small sizes (around 2 nm) and spherical morphologies [15]. The most common method for AuNP synthesis is the citrate reduction of tetrachloroauric (III) acid. After obtaining these NPs, they can easily be functionalized with drugs and/or targeting moieties by physical absorption or by chemical bonding (ionic or covalent) due to the strong binding affinity of thiols, carboxylic acids, disulphides and proteins for Au [16]. In the nineteenth century, Michael Faraday reported the first scientific note on AuNP synthesis. Since then, several synthetic routes and drug conjugations have been proposed. Xiao et al. proposed a conjugated nanosystem aimed at the promotion of human neuroblastoma (SH-SY5Y) cell proliferation and neurite growth. They developed AuNPs with 6-mercaptopurine, an anti-inflammatory drug, exposed on the particle surface through an Au-sulphur bond, and a neuron-penetrating peptide (RDP) [17]. The functionalization with RDP allowed an increased uptake efficiency in the neural cells. These ligands can bring numerous beneficial properties to nanosystems that have already shown very promising results; such is the case of the gold nanoclusters (AuNCs), developed by Gao et al. and composed of AuNPs, which have been reported to have a rare direct medicinal activity [18]. AuNCs inhibited the aggregation and fibrillation of α-Syn. The uncommon aggregation and fibrillation of this synuclein protein, predominantly expressed in neurons’ presynaptic terminals, leads to the formation of Lewy bodies, a remarkable pathological hallmark of Parkinson’s disease (PD) [18]. The promising results provided a proof of principle for the prospect of AuNCs in new drug discovery against PD and the functionalization of these nanoclusters with a targeting moiety (see Section 3) can further confer neurospecificity, improving the effectiveness of this system.

Recently, another type of inorganic nanosystem has been emerging in the drug delivery field—the superparamagnetic iron oxide NPs (SPIONs) or iron oxide NPs (IONPs). These particles are made of small crystals of iron oxide, commonly magnetite (Fe_3_O_4_) or maghemite (γ-Fe_2_O_3_), with attractive nanometric sizes (usually in the range of 20 to 150 nm) [19,20]. These NPs are usually synthesized by coprecipitation and can be coated with different polymers. This strategy, besides improving biocompatibility, allows for an easy functionalization with targeting moieties. 

Many IONPs have been evaluated in preclinical and clinical trials, and several of them are clinically available [19]. Despite there still being a low number of approved iron oxide-based systems for drug delivery to the CNS, interesting nanosystems have been proposed by various research teams and, in the future, could be clinically tested and hopefully approved. Although not being an example of IONPs being used as a drug delivery system, Nanotherm^®^ is a fantastic first approved application in the CNS. In 2010, this iron oxide nanosystem was approved by the Food and Drug Administration (FDA) for the treatment of glioblastomas [21]. Nanotherm^®^ is an innovative therapeutic option based on magnetic hyperthermia where iron oxide particles modified with an aminosilane coating are directly injected into the tumor. Nevertheless, this is an important result that contributes to bringing the use of IONPs closer to the clinic for therapeutics delivery purposes.

Also, focusing on treating diseases in the CNS, Najafabadi et al. developed promising SPION nanosystems conjugated with quercetin, known to protect neuronal cells against oxidative stress and apoptosis [22]. The reported results indicated that the quercetin conjugated to the dextran-coated Fe_3_O_4_ NPs, enhancing the bioavailability of quercetin in the brain and, therefore, could be considered as treatment for neurodegenerative disorders. Besides their size, the magnetic properties of SPIONs can be advantageously applied. Due to their iron oxide core, one approach to target the CNS can be through their magnetic properties. In fact, there are some studies demonstrating that the application of an extracranial magnetic field can mediate the BBB crossing of these NPs, allowing for their accumulation in the brain [23,24]. 

Silica-based or silver-based NPs are other inorganic NPs that have received attention. Silica-based NPs are widely used because their production is easy and inexpensive [25], they present in low nanosizes (less than 50 nm), they have a porous nature (that can greatly increase their surface/volume ratio and, consequently, permit higher surface functionalization capacity) and they do not show swelling or variations in porosity with pH or temperature. Therefore, these particles can successfully protect distinct molecules against denaturation induced by these two latter parameters [26]. Jampilek et al. delivered three nootropics loaded in silica nanosystems. Nootropics are neuroenhancers used after a trauma or as a treatment of neurodegenerative disorders, to enhance memory of other cognitive functions [25]. The site of action of these drugs is the brain and, therefore, the BBB is a serious obstacle when considering systemic injection. However, in this study, it was found that the drug, when carried by silica-based vectors, permeated through the barrier to the brain tissue, increasing the amount of drug in the brain tissue, approximately 200-fold. The combination of two or more types of delivery system has been a very prominent proposal and could be an interesting option for the future. Turan et al. combined silica NPs with an iron oxide core as a treatment for a brain tumor. This inorganic-combination nanosystem allowed for the triggered drug release, by an innovative switchable mechanism, which can be activated by external radiofrequency fields near a brain tumor site [27].

Regarding silver NPs (AgNPs), these have many interesting features, especially their exceptional antimicrobial activity. In fact, CNS treatments for infectious diseases caused by brain-eating amoebae (*Acanthamoeba species*, *Balamuthia mandrillaris*, and *Naegleria fowleri*) have been proposed using these NPs [28]. However, very few studies have been performed using AgNPs as drug delivery systems as a treatment of CNS neurons, especially due to several safety-related aspects [29,30,31]. But these safety issues are not only associated with AgNPs; in fact, in spite of all of the mentioned advantageous properties of inorganic NPs, the majority still have major drawbacks and have been shown to be particularly critical for the brain, as they can induce neurotoxicity, neuroinflammation and can permanently alter the BBB permeability [23,32,33].

##### Lipid-Based Nanosystems

Liposomes, first reported by Alec Bangham in 1961, are the most common and well-investigated nanocarriers for drug delivery [9,34,35]. These spherical vesicles, based on phospholipids, amphiphilic molecules and two fatty acid hydrophobic chains, can enclose aqueous core spaces surrounded by a hydrophobic membrane, and can be prepared with sizes ranging from 30 nm to several micrometers [34,35,36]. Liposomes’ basic properties, such as size, fluidity, rigidity and surface behavior, can be changed by using different lipids and preparation methods. Due to their size, amphipathic character and biocompatibility, liposomes are promising candidates for drug delivery [35]. An alternative to traditional liposomes may be other lipid-based nanocarriers, such as solid lipid NPs (SLNs), nanostructured lipid carriers (NLCs) and lipidic nanobubbles. SLNs are biocolloidal dispersions composed of biocompatible solid lipids, as a particle matrix, prepared by high-pressure homogenization and using mainly polyhydroxy surfactants (sucrose, glucose, or glycerol) as stabilizers. On the other hand, NLCs, which were developed to overcome the limited loading capacity of SLNs, have a matrix composed of solid and liquid lipid blends. Lipidic nanobubbles are a different approach, consisting of gaseous cavities, usually with diameters of <100 nm, that have shown promising results in the neuronal delivery field. Several research groups applied focused ultrasound combined with nanobubbles, accomplishing a reliable BBB opening [37,38]. However, some safety issues were raised due to the collapse of nanobubbles, which can damage membrane proteins, such as proteins in the BBB that play an important role in protecting brain tissue [39]. Because of this, further studies using this type of NP are needed to confirm their safe suitability for neuronal delivery. The amphiphilicity of all these lipidic nanosystems allows the encapsulation of lipophilic and water-soluble drugs, that can be loaded by active (drug encapsulated after liposome/SLN/NLC assembly) or passive (drug encapsulated during assembly) approaches [9]. In addition to this encapsulation capability, lipidic nanosystems can be tuned to release the drug molecules in a controlled and precise way after their functionalization with targeting ligands [34]. The conjugation with these targeting moieties is usually carried out by reaction between the carboxyl group linkage of delivery vector and the functional groups, such as hydroxyl, amine, and thiol group, among others, present in drugs. 

Over the years, advances in the design of the most used lipidic nanocarriers, liposomes, have led to new therapeutic options in the treatment of different conditions, and a number of liposome-based nanosystems have been approved by the FDA and European Drug Agency (EMA) [34,40]. Although most of them are applied as cancer treatments, some of them are indicated for the treatment of meningitis, a pathology characterized by the acute inflammation of the brain and spinal cord protective membranes. Depocyt^®^ is a liposome-based drug approved in U.S. and Europe for the treatment of lymphomatous meningitis and neoplastic meningitis. This may have been the first step in introducing lipidic-based nanosystems as a treatment for CNS diseases. Of note is the fact the first ever approved RNA interference drug by the FDA (Patisiran) is formulated with hepatotropic lipid nanoparticles for the treatment of familial amyloid polyneuropathy [41].

There are some other liposome drug conjugates for the treatment of CNS disorders in different levels of preclinical development to evaluate their pharmacokinetics and biodistribution profiles [34]. Some of these nanosystems have been explored as glioblastoma therapeutics. Several studies have reported the use of liposomal formulations to deliver anti-cancer drugs, such as methotrexate, 5-fluorouracil, paclitaxel, erlotinib and doxorubicin [42,43,44,45,46,47]. Liposomes with sizes smaller than ~200 nm have the capacity to cross the BBB [48,49,50,51]. Wen et al. prepared quantum-dot-loaded liposomes (size <150 nm) that were capable of overcoming the barrier more efficiently than free quantum dots [51]. 

However, to enhance the efficiency of the passage, they have be functionalized with receptor-targeting molecules to enhance the delivery at the targeted site [42,43,52]. Rodrigues et al. modified liposomes with TAT, a cationic peptide derived from the transactivating protein of HIV-1. TAT, known as an enhancer of NP passage through BBB, is one of many cell-penetrating peptides that can be used to functionalize the nanosystem’s surface to improve its internalization capabilities [53]. For further reading on cell penetrating peptides, please refer to the review by Guidotti et al. [54].

Despite all the great advantages related to these vectors, there are some features limiting their applicability to some extent, such as low solubility, poor stability and low encapsulation efficiency, which still need to be overcome [34,55].

##### Polymeric Nanosystems

Polymeric nanosystems could be considered a friendly material for nanomedical applications due to their simplistic synthesis, favorable features, such as biocompatibility and tunable physicochemical properties, and widespread applicability [21]. Innovative polymeric delivery systems have been developed and optimized for diminished toxicity, enhanced bioavailability, biodegradability, and desirable pharmacokinetics. These optimized nanosystems can entrap and protect drug molecules from enzymatic and hydrolytic degradation, mediate their transport and, when the drug is linked via a stimuli-sensitive (e.g., pH, temperature, redox) bond, sustain the drug release within the target site over a period of days or even weeks [26].

Polymers can be synthesized by various mechanisms, such as radical polymerization, condensation polymerization, graft copolymerization, photopolymerization, and ring-opening polymerization [56]. 

The most explored biocompatible synthetic polymeric systems have been poly(glycolic acid) (PGA), poly(ethylene glycol) (PEG), poly(lactic acid) (PLA), poly(ε-caprolactone) (PCL), poly(lactic-co-glycolic acid) (PLGA), poly(amino acids) and N-(2-hydroxypropyl)methacrylamide copolymers (HPMA) [26,57]. In addition to these synthetic polymers, carbohydrate based polymers, such as chitosan and derivatives, as well as poly(cyclodextrins), have also been studied for neuronal delivery [58,59,60,61,62]. 

The biodegradation is a property that can be assured by the existence of labile bonds in their backbone or crosslinker (in the case of reticulated systems). The most commonly explored are the hydrolysable (enzymatic mediated or hydrolysis) ester bonds, such as in the case of the commonly used PLGA, of which the degradation products (lactic acid and glycolic acid) are naturally present in the body and, therefore, it is fully biocompatible [42,43]. PCL is another biodegradable polyester which has a slow degradation process [44]. 

It is noteworthy that PEG is a standard synthetic polymer often used in the development of delivery nanosystems. This polymer can be easily modified at its terminal hydroxyl group with different reactive groups, enabling crosslinking and conjugation chemistries; consequently, it is used in several biomedical applications [63]. Moreover, PEGylation of nanosystems is an approach commonly adopted to increase their water solubility, biocompatibility and circulation time in the bloodstream (by the reduction in opsonization). The use of PEG has also been extensively explored as a linker between targeting moieties and NPs in order to expose the ligand on the NP surface, to coat the NP surface or for both of these purposes at the same time [64].

Recent studies have focused on the use of PLGA for the preparation of NPs to encapsulate therapeutic agents for the treatment of CNS disorders, such as spinal cord injury (SCI), AD or brain cancer [42,65]. Lowry et al. used biodegradable PLGA microspheres to encapsulate sonic hedgehog (Shh), a multifunctional growth factor involved in the proliferation and differentiation of oligodendrocytes and neurons during spinal cord development. They evaluated the efficacy of these NPs as drug delivery systems of Shh at the site of lesion and the consequent functional recovery in the case of SCI [66]. However, the injection of microspheres at the injury site may not be an easy method of administration. Thus, the preparation of particles with smaller sizes and the functionalization with targeting moieties can be beneficial, as it enables administration to be performed in a less invasive way and reach the site of action equally. For instance, Li et al. opted to functionalize PLGA NPs with lactoferrin (Lf) to act as a carrier of shikonin, a potential antiglioma agent, and these were able to pass the BBB and accumulate in the brain after intravenous (IV) injection in the rat tail vein. The accumulation in the brain increased three-fold, compared to free shikonin [67].

In 2013, one of the top 10 best-selling drugs in the U.S. was Copaxone^®^, a polymeric-based drug in which the polymer is glatiramer acetate (a large polypeptide composed of l-glutamic acid, l-alanine, l-lysine and l-tyrosine) [68]. Initially approved in 1996, Copaxone^®^, an immunomodulator that is a result of many years of research, was used in the treatment of multiple sclerosis.

Regarding natural polymers, chitosan has probably been the most extensively explored. Chitosan is a polymer of d-glucosamine and β-1-4 N-acetylglucosamine residues derived by the partial deacetylation of chitin [69]. Bhavna et al. developed chitosan NPs loaded with donepezil, a drug commonly used to treat AD. The intranasal administration of ^14^C-labeled donepezil-loaded NPs in rats resulted in a high percentage of radioactivity per gram in the brain tissue when compared to the drug solution [59,70]. To enhance NP delivery at the neurons, chitosan has also been functionalized with different neurotargeting ligands. For instance, to target the peripheral nervous system neurons (PNS), Lopes et al. selected the non-toxic carboxylic terminal fragment from the tetanus neurotoxin (further discussed in Section 3) as the NP-targeting moiety. These chitosan NPs had great results in rescuing peripheral neurons in vivo from degeneration after intramuscular administration [71].

Despite these and other very promising success stories, several drawbacks may limit the feasibility of this type of NP, such as the toxicity and/or the non-degradability of some cases, whilst the synthetic process can be very complicated, with high costs [60].

##### Dendrimer-Based Nanosystems

Initially, the field of polymer chemistry was mainly focused on linear-shaped molecules. However, innovative structures have emerged which differ from this initial idea. One of the types of compound that revolutionized the polymer chemistry was a group of very branched and globular macromolecules called ‘dendrimers’ [72,73,74]. 

Dendrimer-like structures were first synthesized in 1978 and, during the following decade, Denkewalter, Tomalia, Newkome, and Frechet increased the complexity of these hyperbranched molecules and named them ‘dendrimers’ [74,75]. 

A typical dendrimer is composed by three main parts: the central core, which may be based on one or several functional groups, repeating units or monomers covalently linked to the core organized in branching layers (generations), and the surface with multiple functional groups (or peripheral groups), which play a key role in their physicochemical properties [73,76,77].

Donald Tomalia defined the divergent synthesis by the formation of the dendrimer from the inner multifunctional core extending outward through several repeated coupling reactions. On the other hand, the convergent synthesis is initiated on the perfect branched dendrons (dendrimer wedges), which are then linked to a multifunctional core after the activation/deprotection of their focal point [77]. As a result, the great number of functional groups in the periphery of dendrimers (like thiols, amines, hydroxyl, carboxylic acids and azide groups, among others) allows for the tethering of different bioactive ligands in a specific and controllable manner according to the desired properties and applications, mimicking the multivalency present in several biological systems [78].

Regarding the drug immobilization/load of these vectors, there are two main approaches: by containing the drug encapsulated inside the dendrimer (“dendritic box” model or complexation) or by conjugation through ionic coordination or covalent linkage to the terminal groups of the dendrimer [77]. For targeting, the latter must be the method of choice to link targeting moieties while keeping them exposed to the dendrimer surface.

Thanks to their tunable chemical structure, dendrimers have shown many advantages, such as their high solubility in different media, hydrophobic or hydrophilic cavities in the interior, high biocompatibility, and low immunogenicity [73,76,79,80]. Related to this, and also due to their attractive chemical and physical characteristics, several applications have been presented in the biomedical and materials fields [73,77,79,81].

There are a wide range of different dendrimer families with great potential for gene delivery, drug delivery, and magnetic resonance and computed tomography imaging [73,74,82,83,84]. Among them, the most explored dendrimers as delivery vectors for CNS applications have been the poly(amido amine) (PAMAM) dendrimers [80,85,86], but others, like poly(propylene imine) (PPI) [80,87,88,89,90], poly(ether)-copoly(ester) (PEPE) [91], poly(l-lysine) based (PLL) [92], carbosilane [93], phosphorus [94], and poly(ether imine) (PETIM) [77,95], have also been reported. 

In 2013, Cerqueira et al. proposed a dendritic nanosystem based on a PAMAM core grafted with carboxymethylchitosan to intracellularly deliver the loaded methylprednisolone, a neuroprotective drug used to prevent secondary injury damage following SCI, to glial cells. They assessed the ability to tune the inflammatory reaction after the traumatic injury, specifically by mitigating the response of microglial cells. The NP-treated animals had a reduction in the number of secondary injuries [96]. Moreover, Dhanikula et al. explored PEPE dendrimers as methotrexate carriers for the treatment of gliomas. Their dendritic nanosystems were conjugated with glucosamine, which interacts with the predominant transporter at the mammalian BBB (GLUT1), to enhance passage through the barrier, as well as for tumor targeting. The quantity of methotrexate transported across BBB was three to five times more after loading in the dendrimer. This innovative nanosystem had promising results against glioma cells and avascular human glioma tumor spheroids [91]. Using a different approach, Iezzi et al. conjugated fluocinolone acetonide (a drug to diminish retina neuroinflammation) to a hydroxyl-terminated fourth-generation PAMAM. In this case, sustained drug release was observed for more than 90 days [97]. 

Despite the great potential shown for biological applications, particularly as drug delivery vectors, some limitations have been associated with dendrimers, such as their high cost and complicated synthesis process, and the non-degradable character of the most-used dendrimers, which can lead to bioaccumulation and cytotoxicity, raising safety issues.

### 2.2. The Impact of the Physical and Chemical Properties of the Nanosystems on Their Biological Performance

The physicochemical properties of a nanosystem, such as the size, morphology, hydrophobicity/hydrophilicity, surface functional groups, charge density, and surface properties (Figure 2 and Figure 3), among others, are very important in assuring the efficient transport and release of a therapeutic agent and to ensure the specificity of neuronal targeting. Figure 2B summarizes some of the most relevant physicochemical properties of the nanosystems when envisaging biomedical applications. Different approaches, either during synthesis or post-synthesis, can be followed to tune these properties, allowing a fine design of the delivery vector. For instance, one can confer or improve features such biological stability, which can be crucial to reach neurons after a peripheral administration. Biological stability is a very desirable feature; however, high stability can cause excessive circulation time and undesirable effects. Thus, this property should be properly tuned, depending on the biological application.

Chemical properties are related to the chemical composition and the functional groups present in the NP, the latter being the main key for linking neuronal targeting moieties. Although these can be tethered to the nanosystems through the same strategies as those explored for certain drugs (non-covalent interactions and covalent linkage), the covalent approach (via stable amide, ether, thioether, disulphide, “click” linkages, among others) is the most common one for targeting [99,100]. This approach ensures that the targeting moiety remains linked to the vector under physiological conditions during the whole biological pathway, leading to a more efficient delivery [101]. The type of functional groups present is also crucial for the biological performance of the nanosystem, since they also provide the possibility of modification to improve the NPs’ physicochemical properties. Moreover, the chemical composition and properties ultimately determine and influence other properties like the hydrophobicity, the surface charge and even the size of the nanostructures, in some cases (Figure 2B).

The hydrophilic/hydrophobic balance is also an important parameter to be considered when designing neuronal delivery vectors. Hydrophilic groups, like hydroxyls, amines and carboxylic acids, can be functionalized with small aliphatic moieties to obtain a more hydrophobic nanosystem and to facilitate the passage through the cell membranes, namely BBB. However, an excessive hydrophobicity can result in toxicity, due to their higher tendency to be retained in tissues for longer, extensively metabolized and to excessively interact with membranes, provoking cell lysis. Furthermore, the hydrophobicity of a delivery system also affects its loading capacity and stability, which, consequently, influences biodistribution [102].

The surface charge of a NP can influence its pharmacokinetics, pharmacodynamics, circulation time and biodistribution, since it can determine the fate of NPs when administered in biological systems. Commonly, positively charged NPs are more easily internalized by cells than neutrally and negatively charged NPs, due to their affinity to the negatively charged cellular membrane. However, an excessive charge can cause toxicity issues and low stability in the blood [103]. Thus, a formulation with a balanced surface charge that enables efficient cellular uptake but does not affect cells should be explored at an early stage. Moreover, it is important to underline that NPs’ original surface charges and other characteristics can change when in contact with biological fluids due to the formation of protein corona, altering their fate [104] (point further addressed in Section 5).

The size of the nanosystems is also a determinant parameter for the NP uptake and permeability into the CNS. The nanosize can determine the pharmacokinetics and the pharmacodynamics in the physiological conditions. Although some systems with sizes ≤200 nm can pass through the cells, NPs with sizes ≤100 nm are more likely to cross the BBB and tissues [105,106]. Interestingly, a study from Sonavane et al. using AuNPs with different sizes (15–200 nm) showed the great importance of the size vector. The obtained results showed that, after IV administration in mice, the distribution of AuNPs was dependent on their particle size. Smaller size AuNPs showed a higher distribution in tissues, including the brain, compared to larger AuNPs [107]. Moreover, another interesting report from Pardeshi et al. showed that the usual small size (<200 nm) of SLNs allow them to cross tight endothelial cells of the BBB and escape from the reticuloendothelial system [108]. Different sizes can trigger different biological phenomena and influence the in vivo distribution, targeting-receptor interaction, biological fate, and toxicity of these delivery systems [109].

Interestingly, the shape of the nanosystem can also impact the CNS passage and cellular uptake, and it is affected by the majority of the properties mentioned before. The morphology can have an impact on the ligand targeting–receptor interaction, cellular uptake, transport and degradation. Most of the studies have been focused on spherical NPs, but other morphologies are being explored. Recently, it was proven that non-spherical NPs feature an improved efficacy of delivery since they present a deviation in their hydrodynamic behavior in blood vessels with shear flow. These NPs are able to carry a larger amount of cargo in comparison with classical spherical particles with the same adhesive strength [103].

All of these properties of nanostructures, in sync with biocompatibility and prolonged blood circulation, have been exploited for the development of solutions to allow for the specific delivery of the corresponding drug to the target site, like the CNS.

## 3. Neurotargeting Moieties

As many of the studies reviewed in the previous sections have shown, very few delivery vectors have the ability to reach neurons without neurotargeting moieties. Neurons are cells with unique characteristics, and have a stellate cell body, the perikaryon or soma, with a fine axon and broad dendrites. However, these remarkable features can pose an additional challenge for the delivery of drugs, as they may have to travel long distances to reach the action site. Furthermore, since they are non-dividing cells, this may also be an extra challenge, for instance for gene delivery. Therefore, targeting strategies may be essential to overcome these spare challenges.

The specific delivery of therapeutic molecules to neurons has been a major challenge in the field. Reaching such an aim would certainly improve efficacy while reducing the dose and frequency needed, as well as the off-target and side effects. With the increased knowledge about CNS biology, some of these strategies were inspired by the idea of solving problems by learning from nature and adapting physiological-like cues to specifically target neurons. These bioinspired delivery systems follow natural mechanisms and have been shown to be promising approaches to CNS therapeutics.

In the following sections, the most relevant classes of bioinspired neuron-targeting ligands, such as those based on proteins, protein domains, peptides, antibodies, and nucleic acid-based aptamers, will be discussed (Table 1).

### 3.1. Proteins

In nature, there are several proteins that have the ability to recognize certain receptors in specific cell membranes. Neurons are no exception, and there are a number of proteins that specifically target this cell population and are the focus of this section. Although transferrin (Tf) and other proteins have no neuron tropism, these will be referred to due to their relevance in allowing for the arrival of these nanosystems at the neuronal vicinity. In mediating the nanosystems passing through the CNS barriers and reaching the neural tissue, these may be essential in the development of an efficient CNS delivery vector.

Transferrin receptors (TfR) have been one of the most widely studied protein receptors, as they are overexpressed in the brain capillary endothelial cells [110], as well as in many tumor cells, which renders the Tf family as a useful targeting moiety that enables the BBB transport, or access to brain tumors. The Tf family is a group of iron-binding glycoproteins; however, some homologues exert different functions. A PEG-modified PAMAM dendrimer functionalized with Tf was described to be able to carry nucleic acids to the brain in the aftermath of an IV administration in a PD model [111]. Moreover, AuNPs were functionalized with Tf by Clark et al. to enable the transcytosis across the BBB [112]. These targeted NPs showed an increased capacity to cross an in vitro model of the barrier and enter the mice’s brain parenchyma in great amounts in vivo, after systemic administration (injection via the lateral tail vein).

Lactotransferrin (Lf), a multifunctional protein of the Tf family, was also studied as a ligand in a PAMAM-based delivery vector [113]. Gao and colleagues demonstrated that after an IV injection, the use of Lf-modified PAMAM increased efficiency 2.3-fold, in comparison with the Tf-functionalized PAMAM [6]. Lf is an especially interesting protein, since its receptors, including the low-density lipoprotein (LDL) receptor-related protein-1 (LRP1) and LRP2, are located on neurons (perikaryon, dendrites, axons), the cerebral microvasculature, and, in some cases, glial cells. These receptors are overexpressed in the neurons of patients with PD and it may become the perfect entrance into diseased neurons [114]. Therefore, this protein can be used to functionalize the surface of NPs that will act as treatment of PD. However, the use of Lf is not limited to this application; Lim et al. functionalized PPI dendrimers with Lf for the treatment of glioblastoma. The in vitro and in vivo results, after IV administration, were very promising, showing that this approach can be a strong candidate for treatment [115]. Moreover, Goyal et al. developed new triblock copolymers based on PEG-PLA-PCL conjugated with Lf to act as a brain delivery vector of bacosides, which are used in Indian Ayurveda for the reversal of amnesia. Induced amnesic mice treated with these nanosystems showed significant memory loss reversal [116]. For selective docetaxel delivery into the brain tissue, Singh et al. developed SLN functionalized with Lf. The in vivo results obtained with this nanosystem validated it as a viable vehicle to target drugs to brain [117].

Brain-derived neurotrophic factor (BDNF), a member of the neurotrophin protein family, is another protein that has aroused much interest, as it has the ability to bind to the tropomyosin-related kinase B (TrkB) receptor, with high potency and specificity. This interaction results in the activation of TrkB, which subsequently promotes neuronal survival, differentiation, and synaptic function. Although this protein is not currently being used as a targeting moiety, some derivatives and mimetics have been presented.

Asialoerythropoietin (AEPO) is a glycoprotein that can recognize erythropoietin receptors, expressed on the cellular membranes of neurons and glial cells, and activate them, promoting anti-apoptotic effects. Due to its interesting properties, Fukuta et al. successfully functionalized PEG-liposomes with AEPO. The results obtained in vivo were quite promising, and this formulation may be a hopeful approach to drug development for ischemic stroke therapy [118].

More recently, certain proteins present in viruses, such as the rabies virus glycoprotein (RVG), a protein component of the viral envelope responsible for host cell infection [119], have been exploited due to the possibility of specifically interacting with the nicotinic acetylcholine receptor in neuronal cells [120,121]. This protein has not been used as a targeting moiety, but from this protein, several peptides have been derived and explored, which are reviewed in Section 3.3. In addition to proteins and derived peptides, protein domains have also been studied.

Moreover, there are several microorganisms that have aroused interest because they use specific proteins to act on neurons. These organisms’ great affinity for neurons can be explored as a neurotargeting approach. One such example is the intracellular protozoan parasite, such as *Toxoplasma gondii*, that is able to easily infect neurons. This parasite is estimated to chronically infect the CNS of up to one third of the world’s population [122]. As this is a recent study, its mechanisms of action are still being examined so that a possible treatment can be found. Moreover, the discoveries that will emerge could be used to develop derivatives with applicability as targeting moieties for the CNS neurons.

Neurotoxins, which have potent protein domains, are used by several bacteria to interact with the host organism and have attracted the attention of several research teams in the world. Neurotoxins can act locally, in a peripheral neuron, or distantly, in a CNS neuron, from the infectious site and are responsible for severe diseases. There are already several known toxins that interact with the nervous system [123] that are being investigated for biomedical purposes, such as the cholera toxin b (CTb), botulinum neurotoxins (BoNTs) and the tetanus neurotoxin (TeNT) [123]. The knowledge of their mechanisms of action and structures [123] has allowed researchers to take advantage of their intrinsic potential as targeting moieties. From these toxins, the protein domains of interest can be isolated and applied for biomedical purposes. Some examples of the neurotoxin peptides are discussed in Section 3.3.

Cholera toxin is the main virulence factor in the development of cholera, a highly contagious acute dehydrating diarrheal disease. CTb, with approximately 55 kD, is one of the two major subunits of cholera toxin that provides the specificity in the delivery of the subunit responsible for the disease phenotype (CTa) to target cells. Regarding its specific binding ability to the pentasaccharide moieties of the ganglioside GM1 present in neuron cell membranes [124], the non-toxic pentameric Vibrio cholerae bacteria CTb fragment was investigated as a conjugate of a poly(d-lysine) system. The obtained complexes were shown to successfully target and significantly enhance the transfection of PC12 cells comparing to unmodified complexes [125].

Among the different known neurotoxins, TeNT and BoNTs have been the most studied in recent years. The TeNT is synthesized by *Clostridium tetani* and the toxin is released into the circulation by germinated bacteria in infected tissues, causing spastic paralysis by the blockage of neurotransmitter release from the inhibitory interneurons of the spinal cord [126]. Contrary to the TeNT, BoNTs are a family of bacterial proteins produced by the bacteria *Clostridium botulinum, C. butyricum,* and *C. barati*, which act as potent inhibitors of neurotransmitter release in the peripheral cholinergic nervous system synapses, ensuing flaccid paralysis [126,127,128]. Despite the opposite clinical symptoms of tetanus and botulism, their causative agents have a similar effect in neurons, starting on the peripheral neurons and eventually reaching the central neurons. Both neurotoxins have a three-domain protein structure with a molecular mass of ∼150 kDa [128]. The activated mature toxin consists in a ~50 kDa light chain (LC), which is a zinc protease, and ~100 kDa heavy chain, that encompasses the N-terminal ∼50 kDa translocation domain (HN), and the C-terminal ~50 kDa receptor-binding domain (HC) [129]. The HC domains of both neurotoxins have been used as biological targeting ligands. Andreu et al. modified their cationic liposome-plasmid DNA (pDNA) complexes with these domains [130]. These ligands allowed to increase the transfection in neuronal cells. Moreover, in 2007, Townsend et al. conjugated the HC domain of TeNT to PLGA-PEG NPs, resulting in particles that selectively targeted neuroblastoma cells in vitro [131]. Also, we have explored the tethering of the non-toxic carboxylic fragment HC of TeNT to trimethyl chitosan (TMC) to attain the neuronal targeting of the NPs when administered via a peripheral administration [132]. These NPs were explored as carriers of pDNA encoding for the brain-derived neurotrophic factor (BDNF) in a peripheral nerve injury model, showing that, upon an intramuscular injection, the expression of the therapeutic gene was upregulated both at the peripheral and central level, with neuroprotective and pro-regenerative effects being observed [71]. These results were very promising and highlighted the potential of these HC-NPs as targeted non-viral delivery vectors of therapeutic molecules.

There are also poisons from various animals whose main target is the nervous system. For instance, several snakes, like *Naja naja* and *Vipera palestinae*, have venoms that are protein and polypeptide toxin mixtures that include lethal neurotoxins. Although these neurotoxins are still unknown, there are some toxins from animals that have already being explored, such as Ts1 neurotoxin. This neurotoxin, from the venom of the Brazilian scorpion *Tityus serrulatus,* can target receptors expressed in the membrane of dorsal root ganglion neurons. Carvalho-de-Souza et al. functionalized NPs with this toxin and studied the transfection efficiency in neurons [133]. The neurotargeting potential of this ligand is still being evaluated.

### 3.2. Peptides 

A good alternative to the use of more complex protein moieties are peptides, which consist of natural or synthetic short strings of amino acids (L or D), specifically due to their easy preparation (automated synthesis), small size and stability. In most cases, d-peptides are used due to their higher stability compared to their l-stereoisomers, while maintaining similar chemical and biological properties (side chain topologies and biological activities). Due to these advantageous properties and the ease of obtaining, by reverse inverse isomerization or mirror-image phage display, d-form peptides have been widely used [134].

Deriving from neurotoxins, some peptides have been proposed. Molossin has been studied and evaluated in rat cerebral cortex primary cultures. This peptide is composed of 15 amino acids of the venom of the American pit viper *Crotalus molossus molossus* integrin-targeting domain. Collins et al. functionalized PLL with molossin and this nanosystem showed strong binding to the rat CNS, thus indicating good potential for neuron transfection [135,136]. Furthermore, the CDX peptide derived from candoxin, a novel toxin isolated from the venom of the Malayan krait *Bungarus candidus* that has high binding affinity to nicotinic acetylcholine receptors, showed high binding affinity to these same receptors and has been proven to cross the BBB [104,105]. As the receptors of this peptide are also expressed on neuron membranes, the entry into neurons can occur. CDX-modified liposomal surfaces had significant neuron-targeted delivery in vitro and in vivo, with improved delivery and enhanced therapeutic effect in glioblastoma of the encapsulated doxorubicin [106]. Also derived from a neurotoxin, Tet-1 (a 12 amino acid peptide), which is an analog of tetanus toxin non-virulent HC fragment, has been recognized by phage display [137,138,139]. A Tet1-modified PEI was synthesized and these NPs showed specific and enhanced binding to PC12 and dorsal root ganglion cells [140]. In the posterior work, a similar polymeric system, based on biodegradable cationic chitosan, was employed to investigate the delivery of a neurotrophic factor in a nerve crush injury in vivo model [71,138].

Chlorotoxin (CTX), a 4-kDa peptide isolated from the venom of Israeli scorpion *Leiurus quinquestriatus*, binds specifically to brain glioma cells, making it a promising option as a treatment, especially because this peptide inhibits the migration and the invasion of tumor cells [141]. This effect results in the binding to glioma-specific chloride channels and matrix metalloproteinase-2 (MMP2). Recently, Zhao et al. prepared PEI dendrimers conjugated with PEG, CTX, with a ^131^I radiolabeling, and then entrapped AuNPs. This nanosystem was used for the targeted SPECT/CT imaging and radionuclide therapy of glioma cells in vitro and in vivo using a subcutaneous glioma tumor model. The developed nanosystem demonstrated potential to be applied for glioma targeted diagnosis and therapy [142]. Moreover, Sun et al. developed a nanosystem based on IONPs conjugated to CTX and methotrexate. It demonstrated improved specificity, extended NP retention and increased cytotoxicity toward tumor cells, suggesting that it also possesses potential for applications in cancer diagnosis and treatment [143]. Several peptides have been derived from the previously mentioned RVG using phage display [119,144]. These can bind specifically to the nicotinic acetylcholine receptor, allowing access through the BBB and the spread throughout the brain until reaching the neurons [121]. The RVG29 (29 amino acid peptide) was conjugated to PAMAM- and PEI-based NPs for pDNA delivery. In the case of RVG29-modified PEI NPs, a 1.3-fold increase in gene expression was obtained in vivo regarding non-modified NPs, after IV administration [145]. Using the RVG29-PAMAM NPs, the gene expression assays showed a 2-fold increase in the brain [146]. Moreover, PEGylated TMC modified with RVG peptide carrying Cy5-labeled siRNA to mouse brains led to an increased accumulation when compared to the nontargeted counterpart [147]. These studies have shown the excellent ability RVG bestows on NPs, allowing them to reach the brain, and, in the future, the action on neurons, that also have RVG receptors, can be achieved. 

Another peptide is the BDNF-derived peptide (IKRG), a four-amino acid peptide sequence which has been reported to target TrkB receptors abundantly present in neurons. Xu et al. used this ligand to functionalize their PEG-PCL NPs, significantly improving their selective internalization into neurons of the dorsal root ganglion [148]. The possibility of reaching the CNS neurons makes this peptide a potential option for the development of targeted NPs as a treatment of neurological disorders. Also based on BDNF, LM22A, a mimetic of BDNF, has been studied. Similarly to BDNF, it has the ability to bind to the same highly specific receptors (TrkB); it can also activate them and promote neuronal survival, differentiation, and synaptic function [149]. Presently, the use of LM22A conjugated to delivery vectors has not yet been published. 

Neurotensin (NT) is another promising 13 amino acid peptide that was originally isolated from the bovine hypothalamic neurons, and afterwards it has been identified in neurons of other species. NT was found to undergo quick internalization after receptor binding [150,151]. This peptide was already conjugated to PLL and applied as a delivery system in hemiparkinsonism-induced rats [152,153]. The promising results obtained in this study opened the door for the use of NT as a neurotargeting moiety and, therefore, other teams are evaluating it [154].

Sellers et al. identified a different peptide, the targeted axonal import (TAxI) peptide, which enriched recombinant bacteriophage accumulation and delivered an active enzyme into spinal cord motor neurons in the aftermath of an intramuscular injection in mice [155]. According to the authors, the TAxI peptide showed great potential for clinics, since it can also interact with human spinal cord motor neurons. However, to the best of our knowledge, the use of this peptide conjugated to delivery vehicles has not yet been explored.

For several other CNS pathologies, different promising peptide conjugates have also been introduced, like dendrigraft PLl-PEG-Angiopep for the treatment of PD [156,157]. Angiopep-2, a 19 amino acid peptide, was conjugated to PLL using PEG as linker. Huang and co-workers chose Angiopep-2 as a ligand to lipoprotein receptor-related protein 1 (LRP1), which is highly expressed in a variety of cell types in the brain, including neurons, vascular cells and glial cells [157]. The results obtained in a chronic parkinsonian model upon IV administration were very promising. Later, PAMAM was functionalized, also via a PEG linker with this peptide, which is also overexpressed in brain tumor cells, to obtain a new delivery nanosystem to glial tumors [158]. Moreover, Xin et al. assessed the potential of Angiopep conjugated to PEG-PCL NPs as a brain-targeted drug delivery system. After injection in the mouse caudal vein, the brain coronal section showed a higher accumulation of Angiopep-PEG-PCL NPs in the cortical layer, lateral ventricle, third ventricles and hippocampus than that of non-targeted NPs [159]. For a different purpose, Kadari et al. developed a novel glioblastoma-targeting approach based on SLN functionalized with angiopep-2 functionalized and loaded with docetaxel. The in vivo targeting capability was assessed in a glioblastoma-induced syngeneic mouse model and the results showed the huge potential to deliver therapeutic molecules to glioma cells [160]. Del Grosso et al. also developed angiopep-2-functionalized NPs, but this time for the brain-targeted delivery of enzymes as a replacement therapy in Krabbe disease. The in vivo delivery of enzymes by functionalized PLGA NPs was assessed 4 h after intraperitoneal injection in different organs of CNS and PNS. Moreover, this same team used the PLGA NPs functionalized with a different peptide, the Tf peptide (Tf2). The enzymatic activity was measured in the nervous system demonstrating activity recovery in the brain up to the unaffected mice level. Santi et al. designed Tf binding peptides and Tf2 was the most promising peptide, in terms of binding efficiency and internalization capacity [161]. Tf2 was conjugated to AuNPs and the internalization was effectively promoted. This team is now focused on the further evaluation of this ligand to act as a targeting moiety for the delivery of therapeutics in vitro and in vivo. In a different study, the peptide leptin30 (30 amino acids), which was derived from the polypeptide leptin (146 amino acids), which is secreted into the bloodstream by adipocytes, was linked to dendrigraft PLL through a PEG [92]. Leptin is an endogenic hormone that acts centrally on cells expressing the leptin receptor situated in the hypothalamus and other parts of the brain. Authors proved that leptin30-PLL NPs crossed an in vitro BBB model effectively and accumulated more in the brain after IV administration [92]. Moreover, block copolymers consisting of PEG and PCL were used to link thrombin- or MMP-9-cleavable peptides present in the ischemic brain tissue [162].

Recently, Pep-TGN, which is a new 12 amino acid peptide selected using the phage display peptide library, was identified. When conjugated with PEG-PLGA NPs, it enabled the crossing of the BBB [163]. These NPs were carrying Coumarin-6, which is a fluorescent probe, to show the marker’s accumulation in the brain. The peptide’s promising features will be further explored in regard to CNS therapeutics delivery [163].

### 3.3. Antibodies 

Antibodies are considered great targeting ligands, especially due to their specific behavior in vivo. Consequently, therapeutic monoclonal antibody (mAb) development for the purpose of translation to the clinic continues an active field and, therefore, mAbs prevail as one of the most explored NP targeting ligands.

Several polymeric nanosystems have been surface functionalized with mAb. Chitosan nanospheres were functionalized with PEG bearing the OX26 mAb, which has great affinity for the TfR. Another high affinity mAb for TfR is 8D3, which has already been added to AuNPs [164]. Moreover, Shi et al. linked to liposomes a mAb targeting the TfR, which is present in the BBB and aided the passage of this barrier and arrival to the brain [165]. These new NPs presented good physicochemical properties and great internalization capacity [166]. Although not specific for neurons, this mAb can be very useful for increasing the arrival of nanosystems to the brain. 

Specific moieties are also being explored; for instance, an innovative PLl-based nanosystem coupled to a mAb (MC192) against the neurotrophin receptor p75^NTR^ was developed [167]. Seeburguer et al. reported that the expression of p75^NTR^ is upregulated in spinal motor neurons in patients suffering from amyotrophic lateral sclerosis [168]. Moreover, Barati et al. showed that this delivery system, injected intramuscularly, is internalized following the binding to the receptor and is transported into the neurons of the brain and spinal cord [167]. 

Zhang et al. developed a new promising liposome-based nanosystem targeted with the 83-14 murine mAb to the human insulin receptor at the human glioma cells. The inhibition of cancer cell growth was achieved after IV administration [169]. In another study, 83-14 murine mAb was conjugated to SLN to improve the brain delivery of saquinavir, a protease inhibitor able to hamper the HIV-1 and HIV-2 duplication in their later mature stage. An increase in the concentration of mAb in the NP surface enhanced the percentage permeability across the BBB, and in the in vitro uptake [170].

### 3.4. Aptamers

Aptamers are single stranded oligonucleotide sequences with a stable, defined and easily adapted three-dimensional configuration [171,172]. Similar to antibodies, aptamers have a specific binding capability to in vivo and ex vivo ligands and have attracted attention in the past few years for targeted drug delivery approaches [171,172,173,174,175]. Currently, many developed aptamers can interact with various targets, from simple inorganic molecules to large protein complexes, and even entire cells [176]. When compared to mAbs, aptamers are significantly easier and cheaper to produce and have less non-specific biodistribution and off-target effects [172]. Due to these characteristics, aptamers have naturally been arousing attention in the field of CNS delivery.

Pegaptanib (Macuragen^®^) is an aptamer-based FDA- and European Medicines Agency-approved drug that has been developed to treat macular degeneration using an antiangiogenic strategy by inhibiting the interaction of vascular endothelial growth factor with its receptor [177,178]. This drug was already discontinued, but this is an important to note as, although not neuron targeting, this was the first approved aptamer-based drug that reached the market.

An innovative bioinspired drug delivery nanosystem was developed by conjugating PLGA and the AS1411 aptamer, which is a DNA aptamer capable of specifically binding to nucleolin. Both cancer and endothelial cells highly express this protein in their plasma membrane. The functionalization with the targeting ligand facilitated the delivery of paclitaxel [179]. The mice treated with these loaded-NPs had significantly higher tumor inhibition. These results demonstrated the potential efficacy of AS1411-functionalized NPs for the treatment of gliomas. Also with high affinity for glioma cells, GMT8 has been applied to enhance intracellular drug delivery and spheroid tumor penetration in combination with NPs and docetaxel [180].

Aptamer17 is another developed aptamer that has already shown its ability to distinguish differentiated from undifferentiated PC12 cells [181]. Although there is not yet any translation to targeted delivery systems, this can be explored as a promising targeting moiety specific to the nervous system.

Other aptamers that have shown promising features have been developed to target misfolded or aggregated proteins, such as amyloid-ß, tau, α-synuclein, huntingtin and prion protein, which are often associated with most CNS pathologies [182]. For further reading on aptamers targeting these neuropathies, please refer to the review by Bouvier et al. [182].

## 4. The Intricacies and Challenges of CNS Targeting

As suggested by the previous sections, architecting a NP to reach the CNS and neurons in particular is not a trivial task. Despite the promising developments obtained so far, and the current knowledge of different neurotargeting moieties that can be explored to encompass a specific delivery to different CNS areas, many challenges have to be faced every step of the way to achieve an effective therapy. The difficulty in crossing the CNS barriers, the choice of an adequate and efficient administration route and the demand of achieving an optimal neurospecificity are some of the major concerns in NP design. These topics are further appraised in the next sections, where the pitfalls for each delivery step are described.

### 4.1. Barriers

The CNS barriers are one of the major obstacles in neuronal delivery approaches; therefore, in this section, these challenges will be described in detail and many possibilities to overcome them will be discussed (Figure 3).

#### 4.1.1. Blood-Brain Barrier

The BBB is a dynamic and highly selective interface between the blood and the brain tissue. It constitutes the major obstacle in systemic brain therapies, due to the composing endothelial cells strongly linked to each other by tight junction proteins, where this endothelium is also regulated by astrocytes, pericytes and neurons [183,184]. Therefore, CNS therapeutic research has been more focused on surpassing this barrier.

Many delivery systems for CNS therapies are designed to take advantage of BBB transportation pathways in order to improve their effectiveness [185]. The transport of NPs across the BBB can happen through different pathways: (a) paracellular diffusion; (b) adsorptive mediated transport; (c) carrier-mediated transport; and (d) receptor-mediated transport. The paracellular pathway plays an important role in blood–brain exchanges and is mediated by the tight junctions’ cohesiveness that controls the passage of lipophilic or soluble low molecular weight molecules [186]. Nevertheless, after a systemic administration, the BBB permeation through the referred route may be enhanced by externally inducing a BBB disruption or tuning the NPs’ characteristics. The BBB-induced disruption to enhance brain delivery through the paracellular pathway creates a temporary perturbation of the barrier, which can be achieved by osmotic, chemical (mannitol, bradykinin analogue) or ultrasound disruptions, depending on the technique used [187,188,189]. Also, engineered NPs coated with surfactants are able to stimulate a transient opening of the BBB and increase the diffusion of therapeutic compounds to the brain parenchyma [190]. However, the limited selectivity challenges the efficacy of these approaches (the BBB is compromised so besides the therapeutics other compounds can permeate, including pathogens). Under pathological conditions, as, for example, in stroke, traumatic brain injury, multiple sclerosis, AD and brain tumors, the BBB undergoes significant changes [184,191,192]. As the BBB suffers a disruption caused by tight junction disassembly, this can be seen as a window of opportunity for therapeutic NP delivery through paracellular diffusion [86,193].

The interaction of the NPs with BBB endothelial cells can also be explored as a way to increase their retention in the brain tissue, which can potentiate their permeation to the brain parenchyma. In this case, the therapeutic approach takes advantage of the adsorption-mediated transport. NPs are properly tuned to interact with the brain capillary and promote their permeation to the brain parenchyma by triggering endocytosis or increasing the drug concentration gradient [190,194]. The adsorption capacity to the brain endothelial cells relies on the chemical and biological surface characteristics of the nanocarriers. This phenomenon is facilitated by a lipophilic and/or positively charged surface, which promote the non-specific interaction with the cell membranes. Regarding more specific interactions with microvessel endothelial cells, targeting moieties can be linked to the NPs, such as molecules to target low-density lipoprotein (LDL) [195], insulin [196] and Tf receptors [110]. The receptor-mediated transport shuttles a range of proteins directly to the brain parenchyma by using the vesicular trafficking of the brain endothelial cells (endocytosis, transcytosis and exocytosis) [190,194]. The conjugation of ligands natively recognized by brain endothelial cells, such as Tf [111,112] and Lf [115,116,117], is a promising alternative for brain drug delivery. For further reading on these and other potential ligands, please refer to the review by Lajoie et al. [194]. This described mechanism seems to be one of the most secure and most efficient methods of targeting drugs to the CNS, and it is perceived to be one of the strategies most likely to succeed [197], as demonstrated in previous sections.

Also, the BBB carrier-mediated transporters control the changes of small hydrophilic molecules induced by the linkage to the respective ligands, such as the glucose transporter GLUT1 and the amino acid carrier LAT1 [185,194]. Hereupon, engineered NPs with moieties able to interact with these ligands are prone to permeate the BBB. Glutathione (GSH) has been used as a potential BBB targeting strategy, since BBB endothelial cells express transporters for this molecule. GSH-tagged PEGylated liposomal loaded with doxorubicin has already entered clinical trials for the treatment of brain tumors [198].

Although the creation of a CNS therapy able to cross the BBB is a challenging task, nanotechnology has room for improving and engineering NPs to defy our main brain guard.

#### 4.1.2. Blood-Cerebrospinal Fluid Barrier

The BCSFB consists of the choroid plexus epithelium between the blood and cerebrospinal fluid (CSF) where cells are also linked by tight junctions and the tissue is highly vascularized. Besides the mechanical protection coming from the tight structure, these cells have specialized transporters to maintain the homeostasis and protect the CNS from external pathogen products [199,200,201]. However, when compared to the BBB, this barrier is leakier and NP present in circulation with appropriate sizes can more easily cross it and enter into the CSF. The higher permissiveness of this barrier does not mean a significant increase in substances’ bioavailability in the deep brain parenchyma; therefore, to increase the brain penetration, the vector can be conjugated to brain transporters and enzymes [185,202].

#### 4.1.3. Blood-Spinal Cord Barrier

Additionally, the BSCB is a major player in regulating the spinal cord environment by interfacing this tissue and the blood. The BSCB and the BBB have noticeable differences, even though they share the cellular building blocks. One of the major differences is the decreased expression of the tight junction complex, which explains the slightly increased permeability of the BSCB compared to the BBB [203].

Similarly to the BBB, the transient disruption of the characteristic tight junctions occurring under pathological conditions, for example in SCI, contributes to improving the delivery of NPs to the tissue [204,205,206,207]. Despite the enhanced permeation under pathological conditions, there is a lack of research in surpassing the BSCB using nanotechnology [208]. However, with its analogy to the BBB, it is intuitive that the same principles can be applied for drug delivery systems.

#### 4.1.4. Avascular Arachnoid Barrier

Lastly, the AAB, part of the meningeal covering, enwraps the brain under the dura interfacing the fenestrated dura blood vessels and the CSF [209]. However, it does not represent a significant blood-brain exchange barrier due to its reduced surface area [8].

### 4.2. How to Reach the Brain?

As discussed above, despite all the efforts made so far, only a small percentage of drugs can permeate the BBB [105], and even if a carrier is successfully designed with specific targeting to penetrate this barrier, side effects in off-targeted areas of the brain may emerge. Also, peripheral organ accumulation and fast elimination are verified in certain administration routes and may result in inefficient delivery and a reduction in the therapeutic effect in the CNS. It is noteworthy that the biodistribution of NPs is highly dependent on their intrinsic characteristics, such as the size, shape, hydrophobicity/hydrophilicity and surface properties [210].

Therefore, administration methods should be chosen considering the physicochemical properties of each type of NP in order to achieve higher accumulations in the target site, avoiding some of the potential barriers and possible side effects. In this way, the choice of the delivery method is of great importance to the success of the therapy and many aspects should be taken in consideration. An overview of the most conventionally used methods and some alternative routes of administration for CNS drug delivery are described below.

#### 4.2.1. ‘Conventional’ Administration Routes

When thinking about therapeutics delivery, one idealizes a ‘magic pill’ that is capable of reaching our target successfully after non-invasive administration. Oral administration is a well-known route but, despite offering the greatest compliance to patients, this route has major drawbacks towards an effective drug delivery to the CNS. The harsh environment of the gastrointestinal tract, the first-pass effect, and the fact that therapeutics have to cross the intestinal wall to reach systemic circulation, make the mission of brain delivery almost impossible [211]. Studies with NP formulations, specially using lipidic NPs, have been developed to improve drug delivery to the CNS after oral administration [212,213]. In spite of these improvements, drug adsorption and bioavailability are still very poor [214].

Contrarily to oral delivery, intracerebral, intracerebroventricular (ICV) or intrathecal administrations (injections into the CSF), as well as implants, are the most commonly used methods when referring to CNS drug delivery, but also represent the most invasive strategies. While these administration routes have the advantages of bypassing the BBB, as they can directly and specifically access the target site, they present a severe risk of damaging the surrounding tissues [215]. Furthermore, the low diffusion of the drug into the brain parenchyma limits the area of action and also may result in neurotoxic effects at the administration site due to an exacerbated local drug concentration [216]. It is important to emphasize that there is some controversy about the use of CSF injections as a direct method of drug delivery to the brain parenchyma. Some studies suggest that these routes of administration are more adequate for brain ependyma delivery, as only a small percentage of the CSF drug concentration permeates the brain and only a few millimeters from the surface [202,217]. However, other authors have claimed that intrathecal administration is able to successfully deliver therapeutics to widespread brain areas due to the close communication between the CSF and perivascular spaces [218]. This discussion may be explained by the use of NPs of different natures (e.g., lipid- or polymer-based) and characteristics, which will influence the desired distribution of the therapy by these methods [219].

IV administration is considered the most clinically preferable method of therapeutics administration and is highly employed for NP delivery, as it allows for fast single or repeated administration to every vascularized tissue in a safer way than the methods described before. This route bypasses the absorption barriers of oral administration, meaning that the effect after IV is faster, and in the case of CNS diseases, it can take advantage of the compromised BBB observed in many brain pathologies to reach the brain parenchyma [118]. On the other hand, the ubiquitous distribution of NPs in the blood by IV injection may result in diminished accumulation in the desired action site at the brain and undesired accumulation in peripheral organs [104]. Because of this, designing NPs to avoid this off-targeted delivery is very important. As previously discussed in Section 3, the functionalization of the delivery systems with targeting moieties is one of the most relevant features being explored to overcome this problem, but other aspects, such as NPs’ biocompatibility, charge and hemocompatibility, should also be taken into consideration in IV delivery. It is well established that cationic NPs can better interact with the negatively charged membranes, rather than neutral or anionic NPs, contributing to improved internalization profiles. However, positively charged NPs have short plasma half-lives due to the rapid clearance, which makes the selective delivery of NPs into the brain difficult [103]. Also, with the blood being the first point of contact within the body, many pathophysiological events can be elicited due to the interaction of the NPs with the different blood components. This could result in the activation of the immune system, coagulation or fibrinolysis cascades, hemolysis and protein adsorption [220]. As previously commented, the PEGylation of the delivery agents can increase the circulation time after IV and also help to avoid the immune system and improve biocompatibility [221], but peripheral accumulation and non-selective delivery may still be verified.

#### 4.2.2. ‘Alternative’ Administration Routes

An effective brain delivery technique, capable of surpassing the BBB and the diffusion issue of intracerebral injections discussed in the previous section, is the convection-enhanced delivery (CED). This route is based on a convective bulk-flow process, where drugs are directly infused into the extracellular space with a catheter, taking advantage of the interstitial pathway to attain an homogeneous perfusion throughout the brain [222]. This approach was already successfully used, especially to perform CNS intratumoral infusions of different types of delivery nanosystem, such as IONPs [223] and polymeric NPs [224,225], among others. However, besides being an invasive route of administration, some areas of the brain represent an anatomic barrier for this technique, such as the ependymal or the pial surfaces and, additionally, pathological tissue structural changes may also alter the infusate diffusion. Moreover, drugs that can leak through the BBB or that are rapidly metabolized in the CNS may not be adequate for this type of administration, as they will impair efficient drug delivery [222]. Furthermore, this process requires long periods of time (hours to days) to be efficient and results have shown high variability between experiments [226].

The administration of NPs via intraarterial injections has also been explored as an alternative to IV delivery. As the carotid artery is the major blood supplier to the brain, this delivery method acts like a ‘freeway’, allowing for a higher exposure of the NPs to the area of interest, owing to the first pass through circulation. This method delays the navigation of the delivery vectors throughout the systemic circulation, facilitating an increased accumulation of NPs at the target site and diminishing the rapid systemic clearance verified in IV administration. Intracarotid drug delivery has been widely studied for the administration of chemotherapeutic agents, namely for magnetic NPs, which are known to present short plasma half-lives [227,228]. Additionally, Chertok et al. verified a 30-fold higher accumulation of PEI-modified IONPs in brain tumors with intracarotid administration, when compared to IV [229]. Nevertheless, the intraarterial injection method poses significant risks of embolism or hemorrhages during the intervention [230], and, therefore, the risk-to-benefit ratio needs to be carefully considered.

Intranasal administration to the CNS is gaining growing interest over the oral or systemic routes, as drugs can be delivered in a self and non-invasive fashion directly to the brain. This method allows the delivery of molecules through the trigeminal and olfactory pathways, bypassing the BBB and the BCSFB. The direct absorption of drugs by the nasal epithelial mucosa results in higher bioavailability, better pharmacokinetic profiles and the rapid onset of the therapy [231]. The success of intranasal administration in NP delivery to the brain was already shown in several studies, where the efficacy of this strategy over the IV route was shown [232,233]. Nevertheless, similarly to the other routes of administration, intranasal delivery also has some associated pitfalls. The small administration volume allowed by the nose-to-brain route could be a limiting factor in assuring that sufficient therapeutic drug doses reach the brain and therefore, it is more appropriate for potent drugs. Also, the small epithelial area, short retention time and the mucociliary clearance may reduce therapeutics’ permeation through the nasal mucosa [231]. It should also be highlighted that the majority of studies for CNS delivery found in the open literature are performed using rodent models, and the surface epithelial area of the olfactory regions of rodents is 10 times higher than humans [226], which can hamper the translation of this method to the clinic.

### 4.3. Targeting Ligands Dilemmas in Neurospecific Delivery

If bypassing the barriers is a challenge for CNS therapeutics, targeting a specific pathological neuronal region or type of neuron cell population is also an undeniable struggle in this area. Neurospecific delivery is particularly challenging once neurons are embedded by glial cells in the brain, which have a more phagocytic nature and, therefore, are more prone to internalize therapeutics [188]. However, the increasing knowledge about CNS pathologies at the cellular and molecular level has facilitated the development of active targeted nanosystems, and successful approaches have already been developed for targeting specific CNS cell populations, as previously discussed. Nevertheless, some considerations regarding the choice of the neurospecific strategy have to be carefully pondered.

First of all, finding a receptor with exclusive neurospecific expression is a hard task. For example, the already mentioned TrkB, the receptors for BDNF or its mimetics, are extensively expressed in the CNS, including the cerebral cortex, hippocampus and spinal cord. However, these receptors are also present in non-neuronal cells like the oligodendrocytes [234], or in cells of the peripheral nervous system, which may hamper a neuronal-targeted delivery. Moreover, TrkB is present in a variety of peripheral tissues, like the kidneys and the pancreas, which may lead to unspecific biodistribution profiles [235].

Another important concern is the evaluation of the endogenous ligands to the target receptor and the possibility of competition between them and the chosen targeting moiety. The LRP-1, a receptor that is overexpressed in gliomas and, therefore, has been vastly explored to target brain tumor cells, has more than 30 different ligands of diverse natures (lipoproteins, proteases, virus, toxins) [236]. Also, gangliosides are extensively explored for brain targeting strategies as receptors of toxins and peptide derivatives, like Tet1. However, these receptors, in particular the GT1b ganglioside, are involved in a variety of neurological processes and therefore have a variety of endogenous ligands, such as lectins, growth factors and integrins [237]. Therefore, besides having successful applications, the use of certain targeting moieties must be carefully equated due to the high competition with endogenous ligands, which may hamper the process of the desired target ligand-receptor interaction. A solution could be the development of alternative targeting ligands with an affinity to slightly different epitopes of the receptors, such as the ones developed in the case of TfR (OX26 and 8D3, for example) [238].

The efficacy of the interaction between ligands and receptors and receptor half-life are also important factors to active targeting delivery [236]. The reduction in ligand affinity to its receptor can be observed when attached to NPs. This limitation is usually overcome by the functionalization of NPs with multiple ligand molecules to increase avidity. However, high avidity can also impede NP delivery, as the strength of the interaction is too strong and thus NPs will remain attached to the plasma membrane [98]. One strategy to surpass this drawback is to functionalize NPs with targeting ligands through acid-responsive cleavable linkers, which can dissociate in endosomes and promote the release of NPs from membrane proteins [239].

Also, the determination of the optimal ligand density, allowing maximal cell-specific interaction, is a critical issue in the tailoring of targeted systems. We have previously reported the use of atomic force microscopy (AFM) as a screening tool for the efficient design of targeted NP systems [240]. Exploring force spectroscopy, we were able to obtain new insights into the ligand–receptor mechanism and to determine the optimal targeted NP formulation regarding neuronal cell internalization of PEI complexes with plasmid DNA functionalized with the HC of the tetanus toxin. In line with our findings, Chu et al. investigated the effect of Tet1 ligand density in neuronal cell uptake and concluded that lower densities of ligand resulted in improved internalization of HPMA–oligolysine copolymer NPs [241]. This could be explained by the fact that the excessive targeting ligand density can result in reduced selectivity, as steric hindrances can be formed due to the over proximity between ligands, decreasing their binding ability to neuronal cells and enhancing the binding to non-targeted cells [242]. Hence, it is important to optimize the NPs’ ligand density to enhance targeting efficiency.

Besides the amount of ligand moieties, one aspect that is crucial for the targeting potential of nanosystems is the correct exposure of the targeting ligands. As previously commented, PEGylation is the most popular multifunctional approach for surface coating to both improve physicochemical properties and ensure the correct surface exposure of the corresponding ligands [243]. However, the impact of PEG spacer density and chain length in ligands optimal exposure has been disregarded. Therefore, recently our group studied this issue, showing that higher densities of PEG chains resulted in improved neuronal internalization performances [244]. This is probably due to the “brush-like” conformation that PEG molecules adopt when grafted to a surface at higher densities, allowing a better exposure of the ligands. Also, in the same study, we showed better internalization and delivery efficiencies with NPs grafted with a mixture of longer (5 kDa) and shorter (2 kDa) PEG chains. These shorter spaced chains resulted in improved ligand mobility, in this way enhancing interactions with the host receptor [244]. Thus, the fine-tuning of PEG functionalization is also of great importance for neurospecific delivery.

Other aspects that have also been neglected when designing active targeted NPs for CNS therapeutics are disease progression and aging [105]. The overexpression of selected targets, either on the BBB or in neuronal cells in the case of neuronal diseases, is the most commonly used strategy for targeting delivery. However, the expression of receptors can be modulated throughout the pathological events or with aging. Osgood et al. confirmed that LRP-1 expression was significantly reduced in a rat model of aging [245]. Thus, nanosystems designed to target this BBB- and glioma cell-associated receptor might not be efficient when administered to senior patients. Moreover, receptor regulation is disease specific and is far from being well known, posing an even tougher challenge for drug delivery strategies. Therefore, methods to evaluate and validate the effective NPs’ neurospecificity in biological models that can mimic CNS diseases are required. An innovative strategy using AFM was proposed by Gomes et al. to face this problem, where NPs were attached to the AFM tip and probed against models of different complexities [246]. This molecular recognition force spectroscopy model is a fast screening tool that can be used to identify and study potential targeting agents. But, all in all, further investigations regarding neurospecific nanotechnologies for CNS therapeutics and their dynamic behavior are needed to better understand therapeutics opportunities.

## 5. Concluding Remarks and Future Perspectives

For decades, researchers have been struggling to find safe, controllable and efficient delivery vectors to tackle the alarming and prevailing problem of CNS diseases. The development of ‘smart’ strategies that can incorporate and deliver therapeutics in an effective way and, at the same time, can display the ligand moieties so that they exert their targeting functions in a specific manner, is a feasible answer. The nanotechnology field has made considerable progress and non-viral nanosystems have been proposed and intensively explored for this purpose as an alternative to the hazardous viral vectors. Innovative and multifunctional engineered NPs have already proven their efficacy in reaching the brain by the incorporation of active targeting moieties. Some of the developed nanostrategies are, in fact, inspired by the intrinsic neurospecificity of some viruses, and others were instigated in physiological cues to attain a safe and efficient CNS delivery. Despite the abundance of different vectors for neuronal targeted drug delivery, polymers and dendrimers have been among the most explored, due to the possibility of versatile chemical design and functionalization.

When developing strategies for neurotargeting, the vector of choice needs to overcome multiple cellular barriers, with the BBB representing the main hurdle. Besides this, targeting the exact site of action at the brain represents a crucial feature to avoid the common toxicity and low therapeutic response observed in unselective distributions. Dual-targeted systems are an attractive strategy to surpass the different barriers sequentially and achieve a specific delivery. Several studies have already proved the potential of this approach to different CNS diseases [247,248]. Nevertheless, a problem that has been neglected in NP design is brain diffusion. Once in the brain, NPs have to diffuse through the negatively charged and narrow extracellular spaces to reach their target, simultaneously avoiding brain clearance mechanisms. It is noteworthy that in the context of some neurologic diseases, the inflammatory process makes the diffusion process more difficult, as microglia cells are activated and increase in number, diminishing the already reduced extracellular spaces [249]. Therefore, nanocarriers need to display stability in the biological milieu and many factors such as size, molecular weight, morphology, hydrophobicity, surface composition and properties can alter NPs’ fates [103]. Moreover, the suitable choice of the chemical bonds to link the targeting moiety and the carried drug may also influence the success of drug delivery (Figure 2A).

Regarding NP engineering, special attention must also be paid to the possible alterations to their physicochemical properties upon interaction with biological fluids and even after arrival in the brain. Depending on NPs’ composition, size, shape, hydrophobicity/hydrophilicity balance and surface properties (specially charge), hundreds of types of serum protein can rapidly adsorb to the NPs when in contact with the bloodstream, forming the known ‘protein corona’ [250]. This protein coating changes the surface properties and may induce NP aggregation, inducing a faster clearance by phagocytic and immune cells and facilitating the non-specific uptake of the NPs by random cell types [221]. Also, this shell could influence ligand–receptor interactions, reducing NPs’ targeting skills [250]. Importantly, the formation of protein corona can induce the alteration of the particle zeta potential to slightly negative values, which can also alter NP biodistribution, interaction with BBB and hamper the uptake and the intracellular trafficking in the target cells, impeding a successful therapy [104]. In addition to the already difficult task of predicting the protein corona effects in NPs’ fates, this biomolecular shell is dynamic and its composition is modulated upon passage through the BBB [251]. Moreover, the composition of protein corona can be influenced by the administration route and, also important, it will be disease-specific, which makes the prediction of NPs’ behavior even more challenging [252]. Therefore, carefully designed models to study the influences of protein corona in NP-cell interactions are urgently needed.

In fact, improved and more realistic models are needed to encompass the study of NPs’ CNS targeting formulations in all stages. First of all, relevant models to mimic BBB dynamics would be essential to predict the optimal time of administration in brain diseases that induce BBB opening. For instance, in the case of stroke, a discussion has been raised about the BBB disruption: mono- or biphasic, and for how long and when it occurs. Knowing the exact mechanisms of BBB opening and its pathological changes through disease progression would be crucial in deciding the best therapeutic window of opportunity. Also, careful attention has to be paid to the choice of the specific disease model. Different outcomes were observed between the transient and the permanent ischemic models [253], showing the possible bias of the results for the same disease. Furthermore, variations between the in vitro and in vivo environments may lead to contradictive results and the extrapolation of these results to humans is risky. In fact, this is one of the factors that most contributes to the gap between research and clinical NP application. With the aim of overcoming this gap, 3D models that better mimic the in vivo microenvironment and the interactions between different tissues or cells have been developed in recent years, such as organoids or organ-on-a-chip devices. These models can be manipulated to simulate different CNS diseases and, in this way, they can help to better predict NPs’ effectiveness and biocompatibility [254,255]. Moreover, these models can be used to study disease progression and predict, for example, overexpressed receptors to be explored as targets, or the best time for the success of the strategy.

In addition to this, the uncertain neurotoxicity of nanomaterials induced by the significant accumulation in tissues has hampered nanotechnologies’ regulatory approval and commercialization. For example, magnetic NPs have already shown successful results, either for drug delivery, cancer therapy or imaging purposes. However, their translation to the clinic has been hampered by the immense controversies regarding their biological safety [256]. Therefore, the use of biocompatible and biodegradable materials could be an answer to avoid safety concerns and exacerbated bioaccumulation, and many groups are now focused in the development of this type of carriers. Recently, we reported a new family of biodegradable PEG-GATGE (gallic acid triethylene glycol ester) dendritic block copolymers, as well as their function as siRNA vectors, which showed an adequate degradation rate [78]. Additionally, these dendritic structures allow an easy and versatile functionalization by ‘click’ chemistry, opening a broad range of functionalization possibilities for these vectors. In fact, regarding the carriers’ development, new synthetic ‘green’ and orthogonal strategies, such as the mentioned ‘click’ chemistry, are being explored.

Finally, it is noteworthy that despite the success and versatility of PEG, some literature reports have highlighted the presence of antibodies that specifically recognize this polymer (anti-PEG Abs). This has been correlated with the therapeutic efficacy loss and the increase in therapeutics’ side effects [63]. Therefore, different anti-biofouling polymers are being explored, such as poly(2-oxazoline)s and poly(glycerol) [257].

Despite all these hurdles, CNS-targeted nanotherapeutics hold tremendous potential to be used as treatments of these complex diseases. So, further efforts should be combined to achieve effective and patient-friendly therapies. Based on the discussed hurdles, it can also be expected that the “ideal” carrier of therapeutic agents to the CNS will have to combine, in one entity, several bio-physicochemical features, including multiple targeting moieties, to efficiently deliver its cargo at the intended site, taking into consideration the secluded nature of this tissue.

## Figures and Tables

**Figure 1 pharmaceutics-12-00192-f001:**
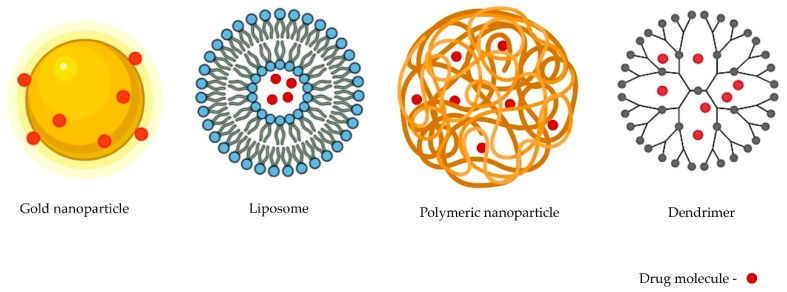
Schematic representation of nanoparticles based on inorganic, lipid, polymer and dendrimer nanostructures applied as therapeutics delivery vectors. Parts of the figure were made with BioRender.

**Figure 2 pharmaceutics-12-00192-f002:**
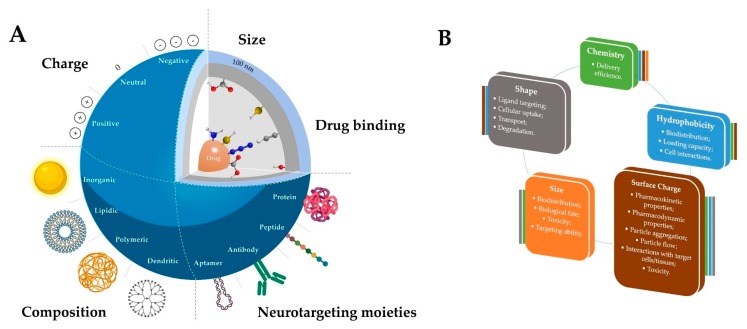
(**A**). Main features of the nanosystems influencing delivery efficiency. Nanoparticles (NPs) can be classified into inorganic (e.g., gold, silica, silver), lipidic, polymeric and dendritic. NPs can be tuned regarding their size (preferably under 100 nm for CNS applications) and are able to bind drugs by the establishment of different bonds/interactions between the functional groups of the drugs and the vector (the atoms were represented according to the standard CPK coloring rules). The NP surface charge can be positive, negative or neutral. Additionally, NPs can be functionalized with different types of targeting ligand, such as aptamers, antibodies, peptides and proteins. The impact of these features on NP performance are further explored in this review. Adapted with permission from: Saraiva et al., Journal of Controlled Release; published by Elsevier, 4773020068925 [98]. Parts of the figure were made with BioRender. (**B**). The influence among the main physicochemical properties of nanosystems and principal biological factors. The color bars represent the interrelationship between the different properties.

**Figure 3 pharmaceutics-12-00192-f003:**
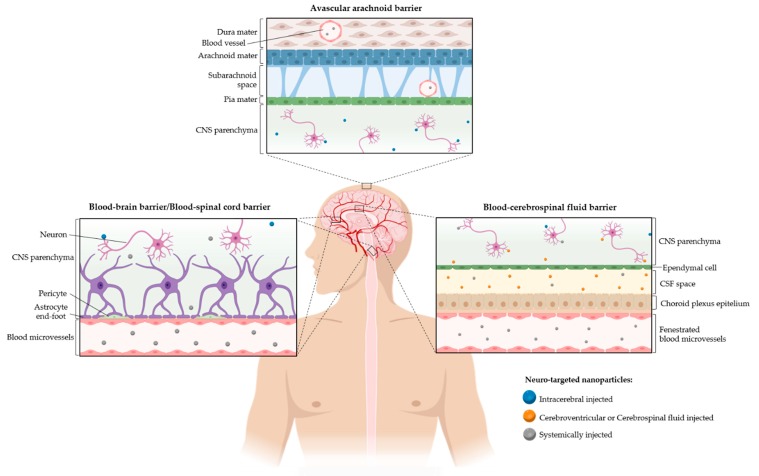
Nanoparticle passage through the CNS barriers as a function of their administration route (intracerebral, cerebroventricular or cerebrospinal and systemic). The figure was created with BioRender.

**Table 1 pharmaceutics-12-00192-t001:** Promising neurotargeting moieties and delivery vectors.

Nature	Targeting Moiety	MW	Associated Vectors	Receptors	Reference
Protein	Lf	80 kD	PAMAM	Lf receptors	[115]
			PPI	LRP1	[117]
			PEG-PLA-PCL	LRP2	[118]
			SLN		[119]
			Liposome		[124]
	BDNF	28 kD	n.a.	TrkB	[120]
	AEPO	~30 kD	Liposome	Erythropoietin receptors	
				nAChR	
	RVG	65 kD	n.a.		
Protein domain	CTb	55 kD	PDL	GM1	[125]
	TeNT	50 kDa	Liposome	GT1b and	[126]
			PLGA	SV2	[127]
			TMC		[128]
			PEI		[129]
	BoNT	50 kDa	Liposome	SV2 and synaptotagmin	[126]
				n.k.	
	Ts1	8 kDa	AuNP		[130]
Peptides	Molossin	2 kDa	PLL	Integrin	[131]
	CDX	29 kDa	Liposome	nAChR	[132]
	Tet1	1.5 kDa	PEI	GT1b	[133]
			Chitosan		[73,134]
	RVG29	3.3 kDa	PAMAM	nAChR	[135]
			PEI		[136]
			TMC		[137]
	IKRG	0.5 kDa	PCL	TrkB	[138]
	LM22A-1	0.5 kDa	n.a.	TrkB	
	NT	1.7 kDa	PLL	NTR-1	[139,140]
	TaxI	1.4 kDa	n.a.	n.k.	
	Angiopep-2	2.3 kDa	PLL	LRP1	[141]
			PAMAM		[142,143]
			PEG-PCL		[144]
			SLN		[145]
			PLGA		[146,147]
			Liposome		[148]
			AuNPs		[149]
	Tf2	1.2 kDa	PLGA	TfR	[147]
			AuNPs		[150]
	Leptin30	3.6 kDa	PLL	ObR	[94]
	Pep-TGN	1.3 kDa	PLGA	n.k.	[151]
	CTX	4 kDa	PEI dendrimer	Chloride channels and MMP2	[152]
			IONPs		[153]
Antibodies	OX26	85–95 kDa	Chitosan	TfR	[154]
			Liposome		[155]
	8D3	22 kDa	AuNP	TfR	[156]
	MC192	75 kDa	PLL	p75^NTR^	[157]
	83-14 murine	~60 kDa	Liposome	Insulin receptor	[158]
			SLN		[159]
Aptamers	AS1411	~8 kDa	PLGA	Nucleolin	[160]
	GMT8	n.k.	PCL	n.k.	[161]
	Aptamer17	26 kDa	n.a.	n.k.	

Lf, lactoferrin, BDNF, brain-derived neurotrophic factor, AEPO, asialoerythropoietin, RVG, rabies virus glycoprotein, CTb, cholera toxin b, TeNT tetanus neurotoxin, BoNT, botulinum neurotoxin, NT, neurotensin, TaxI targeted axonal import, AuNP, gold nanoparticle, Tf2, transferrin peptide 2, CTX, chlorotoxin, PAMAM, poly(amido amine) dendrimer, PPI, poly(propylene imine), PEG-PLA-PCL, polyethylene glycol-polylactic acid-polycaprolactone, SLN, solid lipid nanoparticle, PDL, poly(d-lysine), PLGA, poly(lactic-co-glycolic acid), TMC trimethyl chitosan, PEI, poly(ethylene imine), AuNP, gold nanoparticle, PLL, poly (l-Lysine), PCL Poly(ε-caprolactone), PEG-PCL, polyethylene glycol-polycaprolactone, IONPs, iron oxide nanoparticles, LRP1, lipoprotein receptor-related protein-1, LRP2, lipoprotein receptor-related protein-2, TrkB, tropomyosin-related kinase B, nAChR, nicotinic acetylcholine receptor, NTR-1 neurotensin receptor 1, ObR, astrocyte leptin receptor, TfR, transferrin receptors, MMP2, matrix metalloproteinase-2, n.a., not applied yet, n.k., not known (to the best of our knowledge).

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
