# Peer review of "Breaking Barriers: Bioinspired Strategies for Targeted Neuronal Delivery to the Central Nervous System"

_pharmaceutics, 2020, doi:10.3390/pharmaceutics12020192_

Round 1

Reviewer 1 Report

In their review about using nanoparticles for CNS drug delivery, Spencer et al. describe the different types of nanoparticles used and their relative successes in treating CNS diseases. There are several issues which can make this a stronger manuscript if addressed.

The English is better in some sections than in others. It reads as if different people wrote different sections and combined them without fully proofreading it. Words or phrases in some areas are awkward and not what a fluent English speaker would write. The entire manuscript would benefit from a fluent English speaker proofreading every section. The sentence starting in line 829 with "Neurospecific delivery..." is one of many examples of this.

There is also some discussion about potential peptides or proteins which may improve targeted NP delivery if they are coated on the outside surface. Binding of a peptide to a receptor does not show that a NP coated with this peptide would be an effective drug delivery device. Perhaps sticking to studies that show these peptide-NP conjugates actually deliver drug would be better than just commenting on peptides that bind to neuronal receptors and may potentially work. Lines 570-574 are one example of this but there are several others in the text and tables.

The authors claim in section 4.1.4 that the AAB seals extracellular fluids of the CNS from the rest of the body. Several recent papers showing extracellular fluid drainage from the CNS through the meningeal lymphatics would suggest otherwise. The authors suggest that injection into the CSF would only result in a few milimeters penetration into the brain but many recent studies suggest that intrathecal infusion (at least in rodents) is able to successfully deliver therapeutics to widespread brain areas along perivascular spaces. There seems to be a misunderstanding or misrepresentation in this manuscript with regards to extracellular fluid flow and distribution within the CNS. A better description of extracellular fluid dynamics within the CNS would be helpful as it directly pertains to drug delivery.

The authors also claim that the EPR effect has been "verified" in tumors. In fact, the EPR effect is controversial and has not been “verified” in tumors. Delivering drugs to brain tumors would be simple if it was. A better description of the EPR effect and its controversies is warranted.

Reviewer 2 Report

In this review, authors carefully reviewed the development of central nervous system targeted drug delivery. Authors reviewed the barriers and strategies in this area. Most importantly, authors discussed the challenges in targeted drgu delivery to CNS. Overall, it is a comprehensive and well-organized review. I recommend accept it after minor revision.

Peptides have been widely used as targeting moieties. Recently, D-form peptides are utilized in brain targeted drug delivery due to the good stability and other superiorities, authors are suggested to discuss this issue in section 3.2. In section 4.3, authors discussed that the efficacy of the interaction between ligand-receptor and receptor half-life are also important factors to active targeting delivery. It is good, and there is another strategy to overcome the high binding affinity between ligand and receptor. The strategy is to conjugate ligand with nanoparticles through acidic-cleavable ligand, which could disassociate in endosome and promote release of nanoparticles from membrane proteins and improve exocytosis (ADVANCED FUNCTIONAL MATERIALS, 2018, 28; 180222730). The protein corona could also influence the interaction between ligand and receptor, thus influencing brain targeted drug delivery (Int J Pharm, 2018, 552(1-2), 328-339; Int J Pharm, 2018, 538(1-2), 105-111), while the protein corona was influenced by particle size, charge and ligand modification. Authors should discuss this issue.

Reviewer 3 Report

Review presents nano-enabled strategies for brain delivery. A major concern is that the review does not really add to what is already published and does not really summarise in depth up-to-date current literature in a systematic way for the reader. Thus, it reads as a literature review for new researchers in the field without offering in depth understanding of the challenges. 

Please see specific comments below:

Abstract feels generic and is uninformative Introduction: Quantify impact of neurological disorders currently worldwide and provide latest statistics about predicted impact financially but also in terms of morbidity and mortality.  Line 47: “CNS barriers” – this is unclear – please rephrase Line 53-54: Define these tunable characteristics? What would be ideal targets for these characteristics and lessons learned that can aid nanomedicine design? These are discussed in length in 2.2. but really there is not enough specificity for brain targeted nano-enabled therapies. Section 2.2. is mostly generic and not individualised or brought into life for brain delivery. Defining the barriers is part of the review (4.2) although this is not explicitly stated in the abstract or title? Also it feels that overcoming all of these barriers is critical for brain delivery which is not necessarily the case as some of these will have significantly a higher likelihood to elicit effective pharmacological concentrations of drugs/biotherapeutics. Section 2 extends length of review with limited value. It is written in a generic way and not really focused on nano-enabled technologies for brain delivery. Secondly, it does not focus on nanomedicines with clinical approval either or currently in clinical trials. Additionally, from lipid particles only liposomes are considered but not solid lipid nanoparticles, lipidic bubbles (although microparticulate). Similarly from inorganic particles only gold is considered/visualised although iron oxide is cilinically approved . Please explain rationale for selection of these. Please re-write this section and shorten significantly as currently offers limited information as to how the systems presented should be ideally be designed for brain delivery. This whole section would merit for detailed described tables where studies can be summarised in detail for reader and a short text description provided (presenting w/w composition, size, functionalisation, drug loading, in vitro/in vivo benefits). Please also remove historical development of particles as this is widely known. Figure 2: Please correct “Drug bounding”. Please also change “neurotargeting” to targeting. Schematic although visibly pleasing does not equate clear description in text of ideal characteristics of particles for brain delivery. It also is misleading i.e. 200 nm size is included in the design as there is very little (if any) likelihood for particles of these size to cross the BBB. Please address the ideal characteristics in the main text. “Liposomes have the capacity to cross the BBB” – Please quantify and reference and describe exactly what size liposomes are able to cross. Polymeric nanoparticles section again focus on some of the used polymers not including the greater range of carbohydrate based polymeric nanoparticles (chitosan, chitosan amphiphiles, cyclodextrins, amino acid based polymeric nanoparticles). Similarly a handful of studies are showcased and not really described in detail with little added benefit to reader. Figure 2 and 3 could be merged and again they offer little value Log P can be calculated for a chemical but not for particulates (or nanoplatforms?). The hydrophilic lipophilic balance is a better parameter Charge is discussed again generally with no literature specifically to studies for permeability across the BBB. Although efforts are taken to describe moieties that are targeting neurons, little is offered specifically for permeability across the BBB. Title implies delivery to CNS and not specifically only to neurons. Both need to be described and summarised. You refer to them but you mask their role as proteins to allow arrival of nanosystems to the neuronal vicinity? These needs to be clearly and systematically classified to those that enable permeation of the CNS barriers as you have described them and those that are able to target neurons. Please cite original research that specifies ability of transferrin to target specific cell populations. Lactoferrin is targeting both transferrin and LDL receptors which is not described in review. Please specify is AEPO was successful in vivo or in vitro. Table 1 would benefit from more detail i.e. w/w composition of carriers, size, density of modifications, drug loaded /targeted, in vitro and in vivo findings. Also this table is really not complete for example Angiopep-2 has been used with dendrimers, polymeric particles, lipidic particles but not reported, similar observation for lactoferrin. Full description of peptide amino acid compositions are needed or at least an ideal of molecular weight of targeting moiety. Ideally also the mechanism of permeation needs to be also added along with receptor/carrier protein targeted. Section 4.1 lacks depth in terms of permeation of nanoparticles and their design features. Also which approaches is more likely to lead to effective concentrations is not discussed. Amounts permeating are described as small but not really stated as numbers, which makes a massive difference in the overall success of a strategy. How much can these nanoparticles increase the permeation? What percentage of dose is recovered in preclinical studies? Administration methods are also not compared in terms of their biodistribution characteristics. This makes section 4.2.1 really basic for a pharmaceutical scientist that is the audience of this journal Please read review carefully for English grammar and syntax. Occasional errors throughout text. Spelling errors occasionally as well i.e. line 198: liposomes, line 429 in Nature

The review would greatly benefit for a more in depth and systematic presentation of information as well better organisation of presented information and succinct writing. At the current state I feel if it offers little to pharmaceutical scientists in the field. Instead of focusing in all areas, possibly focusing only in strategies targeting neurons will probably give this review an edge.

Round 2

Reviewer 3 Report

Review presents nano-enabled strategies for brain delivery targeting neurons. A major concern is that the review does not really add to reviews that are already published and does not really summarise in depth up-to-date current literature in a systematic way for the reader. Thus, it reads as a literature review for new researchers in the field without offering in depth understanding of the challenges for pharmaceutical scientists in the field in a clear and concise authoritative manner.  

Review needs to be written more succinctly as it is extremely long but offers little detail. Using tables to summarise studies and provide adequate detail and depth and provide detail of studies of inorganic and organic nanomedicines will greatly enhance usefulness of the review to readers. At the current state I feel if it offers little to pharmaceutical scientists in the field.

Some minor comments:

Introduction: Please add references regarding stated statistics for morbidity of neurological diseases. Section 2.1 is labelled as “nanosystems for therapeutics delivery” but section only refers to viral vectors. Ensure a subheading is added or it accurately describes content. Table 1: Lf receptors? Lactoferin does not have separate receptors? But is binding to transferrin and LDL receptors? Also again, angiopep-2 data are incomplete. Section 4.1 Please add % of dose of strategies reaching the brain for described strategies.
